# Benzene- and pyridine-incorporated octaphyrins with different coordination modes toward two Pd$^{II}$ centers

Le Liu[1], Zhiwen Hu[1], Fenni Zhang[1], Yang Liu[1], Ling Xu[1], Mingbo Zhou[1], Takayuki Tanaka[2], Atsuhiro Osuka[1] & Jianxin Song [1][✉]

Expanded porphyrins have received considerable attention due to their unique optical, electrochemical and coordination properties. Here, we report benzene- and pyridine-incorporated octaphyrins(1.1.0.0.1.1.0.0), which are synthesized through Suzuki-Miyaura coupling of α,α′-diboryltripyrrane with *m*-dibromobenzene and 2,6-dibromopyridine, respectively, and subsequent oxidation with 2,3-dicyano-5,6-dichlorobenzoquinone. Both octaphyrins are nonaromatic and take on dumbbell structures. Upon treatment with Pd(OOCCH$_3$)$_2$, the benzene-incorporated one gives a C$_i$ symmetric NNNC coordinated bis-Pd$^{II}$ complex but the pyridine incorporated one gives C$_i$ and C$_s$ symmetric NNNC coordinated bis-Pd$^{II}$ complexes along with an NNNN coordinated bis-Pd$^{II}$ complex bearing a transannular C–C bond between the pyrrole α-positions. In addition, these two pyridine-containing NNNC Pd$^{II}$ complexes undergo trifluoroacetic acid-induced clean interconversion.

[1] College of Chemistry and Chemical Engineering, Key Laboratory of Chemical Biology and Traditional Chinese Medicine Research (Ministry of Education of China), Key Laboratory of the Assembly and Application of Organic Functional Molecules of Hunan Province, Hunan Normal University, 410081 Changsha, China. [2] Department of Chemistry, Graduate School of Science, Kyoto University, Sakyo-ku, Kyoto 606-8502, Japan. [✉]email: jxsong@hunnu.edu.cn

n recent years, considerable attention has been focused on expanded porphyrins in light of their attractive optical, electrochemical, and coordination properties[1–12]. Among these, pyridine-incorporated expanded porphyrins possess a unique position, since they showed interesting chemical behaviors arising from the basic pyridine subunits. More than two decades ago Corriu et al. reported the synthesis of benzene-incorporated and pyridine-incorporated amethyrin analogs 1 and 2 via a rational route[13]. Later, several analogous molecules possessing unique functions have been reported. As representative examples, cryptand-like molecule 3 exhibited unique properties such as binding three ethanol molecules and positive cooperativity in binding carboxylic acids[14]. Cyclo[2]pyridine[4]pyrrole 4 and cyclo[3]pyridine[3]pyrrole 5 showed protonation-induced realization of global conjugated networks[15] and supramolecular assembling with dicarboxylic acids[16]. Pyridine-modified rubyrin 6 underwent protonation-induced flipping of the pyridine subunits[17], and cyclo[6]pyridine[6]pyrrole 7 was synthesized as the largest pyridine-incorporating expanded porphyrin that showed characteristic conformational flexibility (Fig. 1)[14–23].

Recently we explored various porphyrinoids by using Suzuki–Miyaura coupling. Reported examples include cyclic porphyrin rings, BODIPY-porphyrin hybrids, and earring porphyrins[24–28]. Despite of these studies, we thought that this coupling strategy could be applied to the synthesis of pyridine-incorporated expanded porphyrins. [36]Octaphyrins (1.1.1.1.1.1.1.1) 8 have been known to bind two metal ions within two semi-porphyrin-like pockets and their bis-$Cu^{II}$ and bis-$Co^{II}$ complexes were demonstrated to undergo quantitative splitting reactions to give the corresponding two metalloporphyrins almost quantitatively[29,30].

Here, we report the synthesis of phenylene-incorporated and pyridine-incorporated octaphyrins that have remained largely unexplored. The presence of pyridine units into the octaphyrin scaffold leads to unexpected metalation modes and reactivity of the resultant metal complexes. Since the pyridine is quite basic it enables preferential protonation and metal coordination. The pyridine-incorporated octaphyrin coordinates $Pd^{II}$ to form $C_i$ and $C_s$ symmetric NNNC bis-$Pd^{II}$ complexes,

and an NNNN coordinated bis-$Pd^{II}$ complex bearing a transannular C–C bond between the pyrrole α-positions. The former two complexes can interconvert upon addition of trifluoroacetic acid (TFA).

## Results

**Synthesis and structural characterization of** 12 **and** 14. 1,3-Phenylene-incorporated and 2,6-pyridylene-incorporated octaphyrins(1.1.0.0.1.1.0.0) 12 and 13 are the target molecules in the present study. The key precursor α,α′-Diboryltripyrrane 9 was prepared by regioselective Ir-catalyzed borylation of 5,10-dimesityltripyrrane (Fig. 2)[28]. Firstly, we tried the synthesis of 1,3-phenylene-incorporating octaphyrin 12 by Suzuki-Miyaura coupling of 9 with m-dibromobenzene and subsequent oxidation with 2,3-dichloro-5,6-dicyano-1,4-benzoquinone (DDQ) but this reaction sequence gave linear precursor 10 in 53% yield. We then examined the cyclization of 10 with 9 under similar coupling and oxidation conditions. The equimolecular reaction of 9 and 10 yielded 12 in 3.8% yield but the yield of 12 could be improved to 8.0% when 1.2 equivalent of 9 was applied. The parent ion peak of 12 was detected at $m/z = 1062.5479$ (calcd for $[C_{76}H_{66}N_6]^+ = 1062.5343$ [M]$^+$). The $^1H$ nuclear magnetic resonance (NMR) spectrum shows a symmetric feature displaying a broad singlet at 13.00 ppm due to the pyrrolic NH protons, two doublets at 7.39 and 7.03 ppm and a singlet at 6.33 ppm due to the pyrrolic β-protons, and only three singlet signals due to the methyl protons of the mesityl groups. Collectively, these chemical shifts indicate a nonaromatic character for 12 (see Supplementary Fig. 4 and Supplementary Table 8). Fortunately, we obtained single crystals of 12 by slow diffusion of isopropanol vapor into its chlorobenzene solution. As shown in Fig. 3a, b, 12 takes a planar dumbbell structure with $C_i$ symmetry, in which the two 1,3-phenylene units are pointing inward, being close to overlapping, and the two tripyrrin units are also pointing inward, forming two hemiporphyrin-like pockets. There is no particular bond-length alteration in the 1,3-phenylene units.

Encouraged by the hemiporphyrin-like pockets, we examined $Pd^{II}$ metalation of 12. A solution containing 12, Pd(OAc)$_2$, and NaOAc in a mixture of chloroform and methanol was refluxed

**Fig. 1 Representative structures of benzene-incorporated or pyridine-incorporated expanded porphyrinoids 1–8. 1**, benzene-incorporated amethyrin analog; **2**, pyridine-incorporated amethyrin analog; **3**, cryptand-like molecule; **4**, cyclo[2]pyridine[4]pyrrole; **5**, cyclo[3]pyridine[3]pyrrole; **6**, pyridine-modified rubyrin; **7**, cyclo[6]pyridine[6]pyrrole; **8**, octaphyrin.

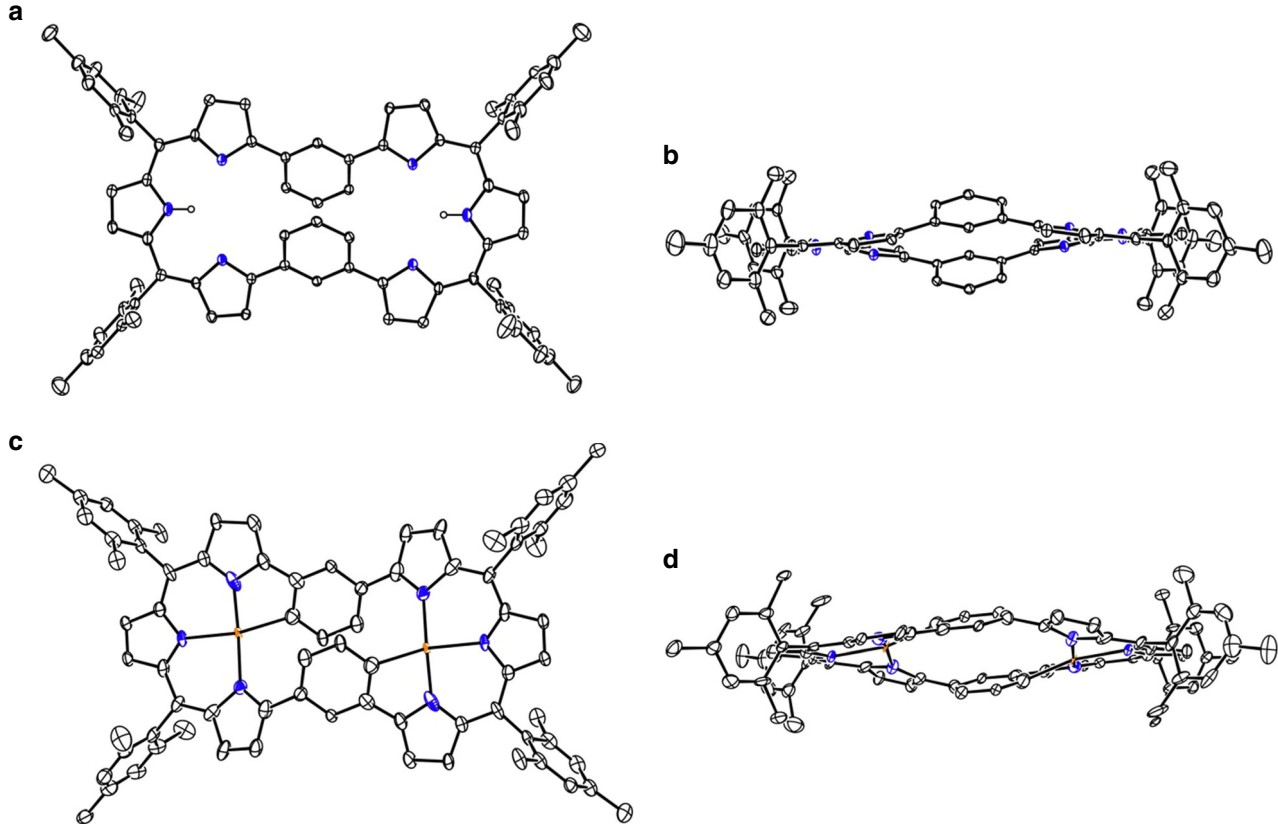

**Fig. 2 Syntheses of 12-17.** Reaction conditions: i) excess *m*-dibromobenzene or 2,6-dibromopyridine, Pd$_2$(dba)$_3$, PPh$_3$, Cs$_2$CO$_3$, CsF, *p*-xylene or toluene/ DMF, reflux, 48 h. ii) DDQ, CH$_2$Cl$_2$, r.t. iii) α,α'-diboryltripyrrane Pd$_2$(dba)$_3$, PPh$_3$, Cs$_2$CO$_3$, CsF, *p*-xylene or toluene/DMF, reflux, 48 h. iv) DDQ, CH$_2$Cl$_2$, r.t. v) *m*-dibromobenzene or 2,6-dibromopyridine, Pd$_2$(dba)$_3$, PPh$_3$ or XPhos, Cs$_2$CO$_3$, CsF, toluene/DMF, reflux, 48 h. vi) DDQ, CH$_2$Cl$_2$, r.t. vii) Pd(OAc)$_2$, NaOAc, CHCl$_3$/CH$_3$OH, reflux. dba = dibenzylideneacetone, Mes = 2,4,6-trimethylphenyl, Bpin = pinacolatoboryl.

**Fig. 3 X-ray structures of 12 and 14. a** Top view and **b** side view of **12**. **c** top view, and **d** side view of **14**. The thermal ellipsoids are on 30% probability level. Hydrogen atoms except those connected to N atoms are omitted for clarity. Carbon atom, black ellipsoid; nitrogen atom, blue; palladium atom, orange; hydrogen atom small black ball.

for 24 h, which yielded **14** in 74% yield as a single product. The parent ion peak was detected at *m/z* = 1270.3018 (calcd for [C$_{76}$H$_{62}$N$_6$Pd$_2$]$^+$ = 1270.3134 [M]$^+$). The structure of **14** was also determined by single crystal X-ray diffraction analysis

(Fig. 3c,d). Complex **14** shows a C$_i$ structure, in which the two Pd$^{II}$ ions take square planar coordination bound with the three pyrrolic nitrogen atoms and one carbon atom in the 1,3-phenylene unit.

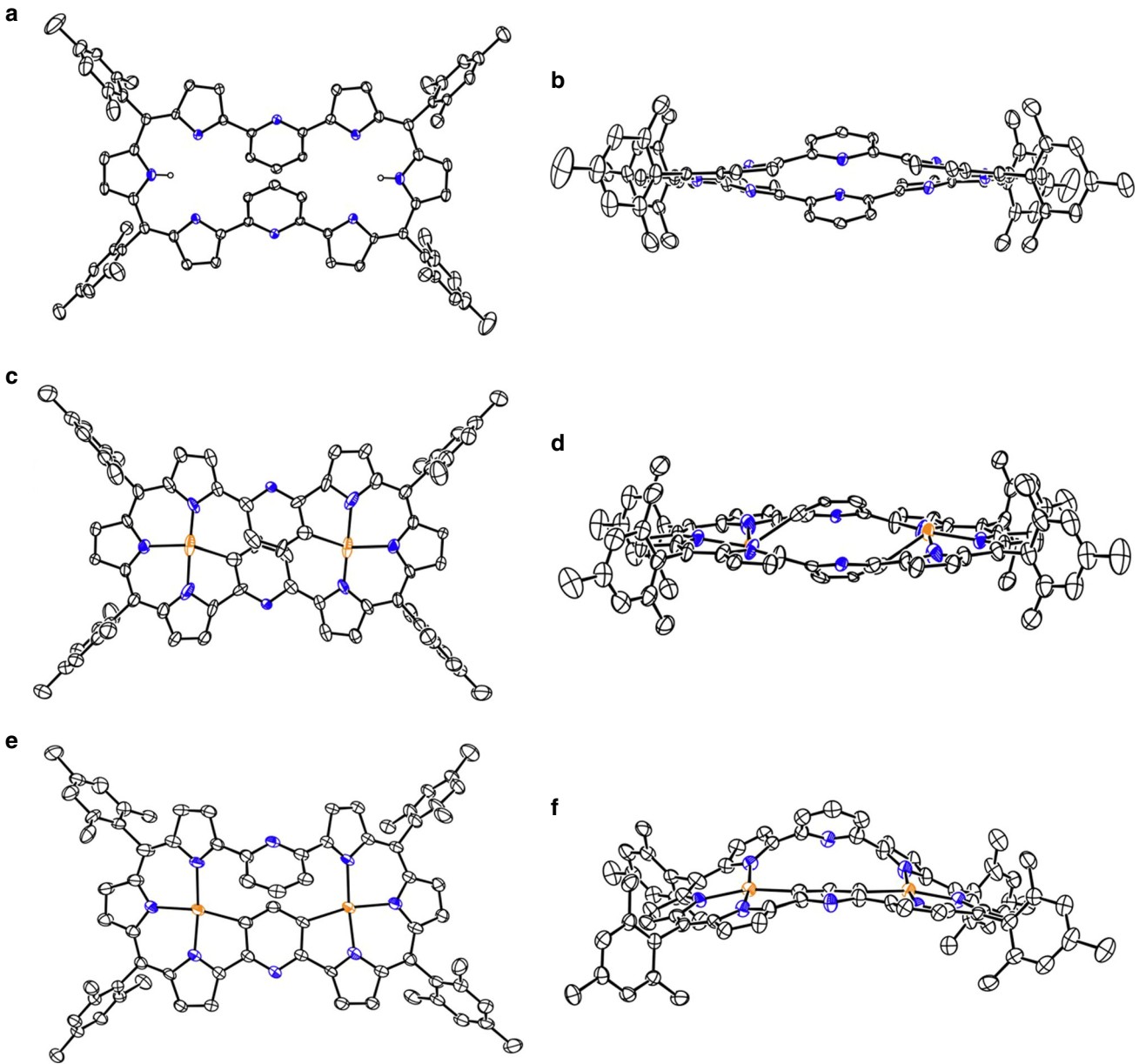

**Fig. 4 X-ray structures of 13, 15, and 16. a** Top view. **b** Side view of **13**. **c** Top view and d) side view of **15**. **e** Top view and **f** side view of **16**. The thermal ellipsoids are on 30% probability level. Hydrogen atoms are omitted for clarity. Carbon atom, black ellipsoid; nitrogen atom, blue; palladium atom, orange; hydrogen atom small black ball.

**Synthesis and structural characterization of** 13, 15, 16, **and** 17. By following the stepwise synthetic protocol for **10**, α,α′-di(6-bromopyrid-2-yl)tripyrrin **11** was prepared by the reaction of **9** with 2,6-dibromopyridine in 48% yield, and octaphyrin **13** was synthesized by the reaction of **9** with **11** in 9.0% yield. The direct one-pot method gave **13** in 8.1% yield. The parent ion peak of **13** was observed at $m/z = 1064.5239$ (calcd for $[C_{74}H_{64}N_8]^+$ 1064.5248 [M]$^+$) and the $^1$H NMR spectrum of **13** is similar to that of **12**, indicating its symmetric structure and a nonaromatic character. Octaphyrin **13** shows a planar dumbbell structure quite similar to that of **12**, in which the two hemiporphyrin-like cavities are secured (Fig. 4a, b).

Considering the high coordination ability of pyridine, it was thought that metalation behaviors of **13** might be different from those of **12**. Pd$^{II}$ metalation of **13** under the same conditions afforded three Pd$^{II}$ complexes **15**, **16**, and **17** in 20%, 60%, and 12% yield, respectively. The structures of **15** and **16** are analogous

to that of **14** but that of **17** is substantially different. The parent ion peaks of **15** and **16** were observed at $m/z = 1272.2944$ (calcd for $[C_{74}H_{60}N_8Pd_2]^+$ 1272.3038 [M]$^+$) and $m/z = 1272.3063$ (calcd for $[C_{74}H_{60}N_8Pd_2]^+$ 1272.3038 [M]$^+$), respectively.

Structures of **15** and **16** were unambiguously determined by X-ray analysis as shown in Fig. 4c–f. Both complexes show the similar planar dumbbell structures but differ in Pd$^{II}$ coordination modes. Namely, in **15** the Pd$^{II}$ metals are respectively bound to different pyridine units, while the two Pd$^{II}$ metals in **16** are bound to the same pyridine unit in a $C_s$ symmetric manner. The Pd–N bond lengths are 1.969(5), 1.978(6), 2.053(6), 2.069(6), 2.071(5), and 2.085(6) Å, and Pd–C bond lengths are 2.077(7) and 2.102(7) Å in **16**. The sum of the bond angles around the Pd ions are 361.4 (3)° and 361.1(3)° for **16**, being closer to the ideal square-planar coordination geometry. In line with these structures, the $^1$H NMR spectrum of **15** shows a set of doublets at 5.46 and 4.07 ppm due to the 2,6-pyridinylene units and that of **16** displays a singlet at

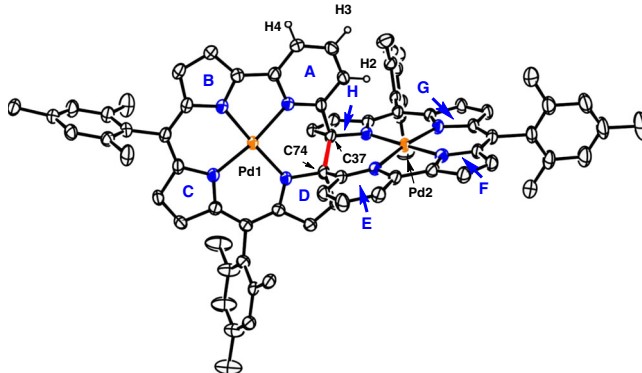

**Fig. 5 X-ray structure: 17.** The thermal ellipsoids are on 30% probability level. Hydrogen atoms except those in 2,6-pyridylene are omitted for clarity. Carbon atom, black ellipsoid; nitrogen atom, blue; palladium atom, orange; hydrogen atom small black ball.

3.69 ppm and a set of mutually coupled doublet at 7.67 ppm and a triplet at 7.03 ppm. These chemical shifts can be explained by the local aromaticity of 1,3-phenylene and 2,6-pyridylene, which meet well with the NICS calculation. DFT calculation reveals that either in **15** or **16** $d$ orbitals of Pd atoms are clearly involved in the HOMO and LUMO orbitals, which indicates a strong electron interaction between Pd atoms and octaphyrin scaffolds.

The structure of **17** is shown in Fig. 5. The Pd1 atom is coordinated with three pyrroles B, C, D and pyridine A in a square planar manner with bond lengths of 1.922(3), 1.962(6), 1.942(4), and 2.024(5) Å. The Pd2 atom is bounded to the three nitrogen atoms of pyrrole rings F, G, H and pyridine E with distances of 1.926(4), 1.971(5), 1.947(4), and 2.023(4) Å. This hemi-porphyrin-like unit is relatively planar with a small mean plane deviation of 0.037(6) Å. Surprisingly, a new C–C bond is formed between the α-positions of the pyrroles D and H, causing a disruption of the macrocyclic conjugated network. This transannular bond length is rather large, being 1.568(8) Å. As a consequence, complex **17** exhibits a roughly perpendicular arrangement of two hemiporphyrin-like units. The parent ion peak was detected at $m/z = 1274.3141$ (calcd for $[C_{74}H_{62}N_8Pd_2]^+$ 1274.3195 $[M]^+$), which is two units larger than those of **15** and **16**. The $^1$H NMR and $^{13}$C NMR spectra are consistent with the structure. Characteristically, the $^{13}$C NMR spectrum shows a signal at 91.44 ppm due to the sp³-hybridized quaternary carbon atoms. The detailed reaction mechanism is not clear but may involve the bis-Pd$^{II}$ complex **18**, which may be highly distorted and undergo a transannular C–C bond forming reaction to give **17** (Fig. 6). This process differs from the previous transannular reactions of expanded porphyrins (Fig. 7)[30–35].

**Electrochemical and optical properties of 12, 13, 14, 15, 16, and 17.** Cyclic voltammetry (CV) and differential-pulse voltammetry (DPV) experiments were conducted and the redox potentials are summarized in Table 1.

In all cases, the oxidation waves are irreversible, indicating certain instabilities under oxidative conditions. On the other hands, the first and second reduction waves are reversible, showing a stepwise 1e⁻ reduction at –1.31 and –1.48 V for **12**, at –1.07 and –1.29 V for **14**. The Pd$^{II}$ complexation result in a definitive anodic shift by about 0.2 V. It is also the case for **13**, the reduction peaks were observed at –0.92 and –1.18 V for **15** and at –0.97 and –1.20 V for **16**, both of which were anodically shifted from those of **13**. Interestingly, the first oxidation potential of **16** (0.47 V) is more negatively shifted than that of **15** (0.62 V), resulting in the smallest gap ($\Delta E_{HL}$) of 1.44 eV. The $\Delta E_{HL}$ values

of **14** and **15** are almost identical, which is consistent with the orders of absorption red-shifts. The first oxidation potential of **17** was observed at –0.02 V, indicating its electron rich nature. This may be accounted for by the presence of two 2-hydropyrrole units in **17**.

The UV-Vis absorption spectra of **12** and **13** are quite similar, exhibiting bands at 402, 551, and 598 nm and at 406, 555, and 610 nm, respectively. The absorption spectrum of **14** is broader and red-shifted, showing bands at 422, 599, and 700 nm. The absorption spectra of **15** and **16** are even broader and red-shifted. The spectral profiles of **15** and **16** are similar in the range of 350–500 nm but are different in the low energy region. Namely, **15** shows a tail up to 850 nm but **16** shows a tail reaching around 1000 nm. The absorption spectrum of **17** is also broader and red-shifted, exhibiting absorption bands at 444, 651, and 697 nm. Upon addition of acid, the absorption spectrum of **13** was largely changed probably due to the protonation of the pyridine moieties, since the addition of acid to $CH_2Cl_2$ solution of **12** did not lead to any changes in its UV–Vis spectrum (Supplementary Fig. 27).

**Mutual interconversion between 15 and 16.** In the meanwhile, we found TFA-triggered clean mutual interconversion between **15** and **16**. In neutral or basic solutions, the mutual interconversion was not detected even at high temperature. Small amounts of TFA (0.05 or 0.2 equiv) accelerated the interconversion with molar ratios of **15/16** at the saturation point: 3.2 or 1.1 (see Supplementary Figs. 40 and 41). On the other hand, in the presence of 1.0 or 2.0 equiv TFA, the interconversion was rapid but gave **16** predominantly (see Supplementary Figs. 42 and 43). After certain reaction time, the interconversion showed the same saturation feature starting from either **15** or **16**, suggesting thermodynamic control. We thought that the relative stability of neutral **15** to **16** determined the equilibrium at low concentration of TFA and that of monoprotonated **15** to **16** determined the equilibrium at high concentration of TFA. TFA-d was then applied to explore the process of proton transferring process. Spectra of the product **16-d** indicated that C–H activation and the following C–Pd bonds deuteration underwent in the isomerization process. To better understand this process DFT calculations of the relative energies of **15** and **16** in neutral and monoprotonated forms were conducted (see Supplementary Figs. 56 and 57). In the monoprotonated forms, the most stable form was **16** protonated at the Pd-coordinated pyridine site, which is 9.0 kcal/mol more stable than **15**, in line with the experimental results. In contrast, the Gibbs free energy difference between neutral **15** and **16** is subtle (~2.7 kcal/mol), so that the equilibrium may be influenced by solvent effects or other experimental conditions.

**Discussion**

In summary, the benzene and pyridine incorporating octaphyrins **12** and **13** were synthesized through Suzuki–Miyaura coupling reaction of the simple starting materials and subsequent oxidation with DDQ. These octaphyrins are nonaromatic and show dumbbell structures. The incorporation of pyridine units into the octaphyrin framework has strong influences such as the formation of the isomeric bis-Pd$^{II}$ complexes **15** and **16** along with the rearranged complex **17**. Further, the complexes **15** and **16** underwent the TFA-induced mutual interconversion depending upon the concentration of TFA. Further exploration of core-modified expanded porphyrins is underway in our laboratories.

**Methods**

**Materials, characterizations, and theoretical calculations.** $^1$H NMR (500 MHz) spectra were measured by a Bruker AVANCE-500 spectrometer, and chemical shifts were reported on the delta scale in ppm relative to CHCl$_3$ as an internal

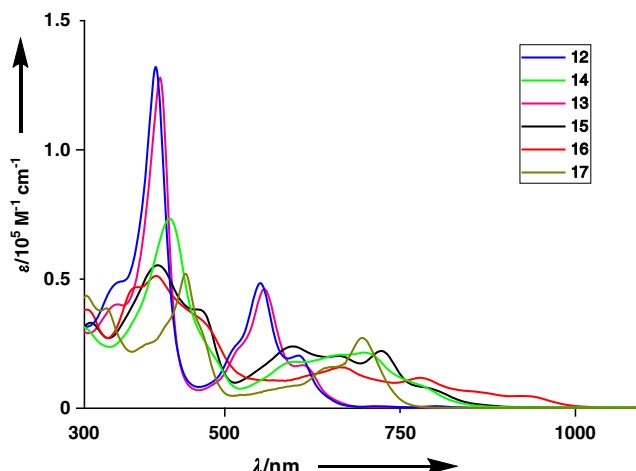

**Fig. 6 Plausible reaction mechanism from 13 to 17. 13** to **13′**, conformation variation via σ bond rotation; **13′** to **18**, metal coordination; **18** to **17**, C–C bond (red) forming.

**Fig. 7 Absorption spectra of 12–17 in CH₂Cl₂.** UV–Vis–NIR absorption spectra of **12** (blue), **14** (green), **13** (purple), **15** (black), **16** (red), and **17** (khaki). λ, wavelength; ε molar extinction coefficient.

**Table 1 Electrochemical properties of 12, 14, 13, 15, 16, and 17 measured in PhCN.**

| Compound | $E_{ox.2}$ | $E_{ox.1}$ | $E_{red.1}$ | $E_{red.2}$ | $E_{red.3}$ | $E_{red.4}$ | $\Delta E_{HL}$ |
|---|---|---|---|---|---|---|---|
| 12 | 0.71[a] | 0.63[a] | −1.31 | −1.48 | −1.80[a] | | 1.94 |
| 14 | 0.74[a] | 0.48[a] | −1.07 | −1.29 | −1.84 | −2.03 | 1.55 |
| 13 | 0.78[a] | 0.70[a] | −1.18 | −1.37 | −1.69[a] | | 1.88 |
| 15 | 0.84[a] | 0.62[a] | −0.92 | −1.18 | −1.74 | −1.94 | 1.54 |
| 16 | 0.94[a] | 0.47[a] | −0.97 | −1.20 | −1.81 | −2.02 | 1.44 |
| 17 | 0.10[a] | −0.02[a] | −1.73 | | | | 1.71 |

Potentials [V] vs. ferrocene/ferrocenium ion. Scan rate 0.05 Vs⁻¹; working electrode, glassy carbon; counter electrode, Pt wire; supporting electrolyte, 0.1 M n-Bu₄NPF₆; reference electrode, Ag/AgNO₃. Electrochemical HOMO-LUMO gaps ($\Delta E_{HL} = E_{ox.1} - E_{red.1}$ [eV]). [a]Irreversible peaks.

reference (δ = 7.260 ppm). Assignments of ¹H NMR were based on HH COSY spectra and D₂O exchange experiments. UV/Vis absorption spectra were recorded on a Shimadzu UV-3600 spectrometer. X-ray crystallographic data were taken on an Agilent SuperNova X-ray diffractometer equipped with a large area CCD detector. Using Olex2, structures of compounds **12–17** were solved with the ShelXS structure solution program using Direct Methods and refined with the ShelXL refinement package using Least Squares minimization. Disordered solvent molecules were treated by SQUEEZE program of Platon. Redox potentials were

measured by the cyclic voltammetry and differential pulse voltammetry method on an ALS660 electrochemical analyzed model (Solvent: PhCN, electrolyte: 0.1 M n-Bu₄NPF₆, working electrode: glassy carbon, reference electrode: Ag/AgNO₃, Counter electrode: Pt wire, scan rate: 0.05 V/s, external reference: ferrocene/ferrocenium cation). Benzonitrile passed through alumina column was used for electrochemical analysis. Unless otherwise noted, materials obtained from commercial suppliers were used without further purification. All calculations were carried out using the Gaussian 09 program[36]. Initial geometries for **12–17** were obtained from X-ray structures. The structures were fully optimized without any symmetry restriction. Geometry optimizations in the ground state (S₀) were performed by the density functional theory (DFT) method with restricted B3LYP (Becke's three-parameter hybrid exchange functionals and the Lee–Yang–Parr correlation functional)[37,38] level employing basis sets and pseudopotentials; 6–311G(d,p) for C, H, N[39], and SDD for Pd[40]. NICS(0) values were calculated with GIAO method at the B3LYP level employing the same basis sets and pseudopotentials for geometry optimizations. Calculated chemical shifts were estimated relative to the magnetic shielding of a proton of chloroform (24.95 ppm) calculated at the same level.

**Synthesis of 10**. A toluene-DMF solution (3/1.5 mL) of **9** (178.4 mg, 0.25 mmol), 1,3-dibromobenzene (301.7 μL, 589.8 mg, 2.5 mmol), Pd₂(dba)₃ (22.9 mg, 0.025 mmol), PPh₃ (26.2 mg, 0.1 mmol), Cs₂CO₃ (172.7 mg, 0.53 mmol), and CsF (77.5 mg, 0.51 mmol) was degassed through three freeze-pump-thaw cycles, and the reaction Schlenk tube was purged with argon. The resulting mixture was stirred at reflux for 48 h. The reaction mixture was diluted with CHCl₃, washed with water, and dried over anhydrous sodium sulfate. After the solvent was removed, 2,3-dichloro-5,6-dicyano-1,4-benzoquinone (DDQ) (136.2 mg, 0.6 mmol) was added to the resulting mixture in CH₂Cl₂ and this reaction mixture was stirred for another 4 h. Evaporation of the solvent followed by silica-gel column chromatography (CH₂Cl₂/n-hexane as an eluent) and recrystallization with CH₃OH to afford **10** as dark green solids (101.7 mg, 0.13 mmol, 53% yield). ¹H NMR (500 MHz) (CDCl₃) δ = 13.60 (br, 1H), 8.00 (s, 2H), 7.68 (d, 2H, J = 7.7 Hz), 7.20 (d, 2H, J = 8.0 Hz), 7.14 (d, 2H, J = 4.5 Hz), 6.97 (s, 4H), 6.88 (d, 2H, J = 4.5 Hz), 6.75 (t, 2H, J = 7.6 Hz), 6.19 (s, 2H), 2.38 (s, 6H), 2.18 (s, 12H) ppm; ¹³C NMR (126 MHz) (CDCl₃): δ = 168.2, 152.7, 139.0, 137.7, 137.3, 137.0, 135.7, 133.6, 132.1, 131.0, 128.8, 127.9, 126.8, 124.1, 122.1, 121.0, 21.1, and 20.5 ppm; HRMS (m/z): [M]⁺ calcd. for C₄₄H₃₇Br₂N₃, 765.1349; found 765.1291.

**Synthesis of 11**. A toluene-DMF solution (3/1.5 mL) of **9** (178.4 mg, 0.25 mmol), 2,6-dibromopyridine (592.2 mg, 2.5 mmol), Pd₂(dba)₃ (22.9 mg, 0.025 mmol), PPh₃ (26.2 mg, 0.1 mmol), Cs₂CO₃ (172.7 mg, 0.53 mmol), and CsF (77.5 mg, 0.51 mmol) was degassed through three freeze-pump-thaw cycles, and the reaction Schlenk tube was purged with argon. The resulting mixture was stirred at reflux for 48 h. The reaction mixture was diluted with CHCl₃, washed with water, and dried over anhydrous sodium sulfate. After the solvent was removed, DDQ (136.2 mg, 0.6 mmol) was added to the resulting mixture in CH₂Cl₂ and this reaction mixture was stirred for another 4 h. The reaction mixture was passed through a short alumina column (CH₂Cl₂ as an eluent) and recrystallization with CH₃OH to afford **11** as dark green solids (91.9 mg, 0.119 mmol, 47.8% yield). ¹H NMR (500 MHz) (CDCl₃): δ = 13.31 (br, 1H), 7.47 (m, 4H), 7.27 (d, 2H), 6.97 (s, 4H), 6.89 (d, 2H, J = 4.5 Hz), 6.66 (t, 2H, J = 7.6 Hz), 6.28 (s, 2H), 2.38 (s, 6H), 2.16 (s, 12H) ppm.

$^{13}$C NMR (126 MHz) (CDCl$_3$): $\delta$ = 168.4, 153.2, 153.0, 140.8, 139.4, 139.0, 138.0, 137.5, 137.1, 135.5, 133.4, 128.0, 127.5, 125.9, 121.9, 121.8, 21.1, 20.5 ppm; HRMS ($m/z$): [M+H]$^+$ calcd. for C$_{42}$H$_{36}$Br$_2$N$_5$, 768.1332; found 768.1354.

**Synthesis of 12**. Route 1: A $p$-xylene-DMF solution (3/1.5 mL) of **9** (84 mg, 0.12 mmol), **10** (77.7 mg, 0.1 mmol), Pd$_2$(dba)$_3$ (18.5 mg, 0.02 mmol), PPh$_3$ (21.2 mg, 0.08 mmol), Cs$_2$CO$_3$ (71.0 mg, 0.2 mmol), and CsF (33.4 mg, 0.2 mmol) was degassed through three freeze-pump-thaw cycles, and the reaction Schlenk tube was purged with argon. The resulting mixture was stirred at reflux for 48 h. The reaction mixture was diluted with CHCl$_3$, washed with water, and dried over anhydrous sodium sulfate. After the solvent was removed, DDQ (65.8 mg, 0.29 mmol) was added to the resulting mixture in CH$_2$Cl$_2$, and this reaction mixture was stirred for another 8 h. The reaction mixture was passed through a short alumina column (CH$_2$Cl$_2$ as an eluent). Evaporation of the solvent followed by silica-gel column chromatography (CH$_2$Cl$_2$/n-hexane as an eluent) and recrystallization with CH$_2$Cl$_2$/MeOH afforded **12** as violet solids (8.4 mg, 0.008 mmol, 8.0% yield).

Route 2: A toluene-DMF solution (3/1.5 mL) of **9** (180 mg, 0.25 mmol), 1,3-dibromobenzene (25.6 μL, 49.5 mg, 0.21 mmol), Pd$_2$(dba)$_3$ (23 mg, 0.025 mmol), X-Phos (48 mg, 0.10 mmol), Cs$_2$CO$_3$ (136.0 mg, 0.42 mmol), and CsF (67.0 mg, 0.44 mmol) was degassed through three freeze-pump-thaw cycles, and the reaction flask was purged with argon. The resulting mixture was stirred for 48 h at reflux. The reaction mixture was diluted with CHCl$_3$, washed with water, and dried over anhydrous sodium sulfate. After the solvent was removed, DDQ (136.2 mg, 0.6 mmol) was added to the resulting mixture in CH$_2$Cl$_2$ and this reaction mixture was stirred for another 8 h. The reaction mixture was passed through a short alumina column (CH$_2$Cl$_2$ as an eluent). Evaporation of the solvent followed by column chromatography on silica gel (CH$_2$Cl$_2$/n-hexane as an eluent) and recrystallization from n-hexane gave **12** as violet solids (7.9 mg, 0.007 mmol, 7.0% yield). $^1$H NMR (500 MHz) (CDCl$_3$): $\delta$ = 13.0 (br, 2H, N–H), 8.34 (br, 2H, $m$-phenylene-H), 7.39 (d, 4H, $J$ = 4.5 Hz, pyrrole-H), 7.25 (dd, 4H, $J$ = 7.8, 1.5 Hz, $m$-phenylene-H), 7.03–7.00 (m, 12H, pyrrole-H and Ar-$m$-H), 6.33 (s, 4H, pyrrole-H), 5.63 (t, 2H, $J$ = 7.7 Hz, $m$-phenylene-H), 2.40 (s, 12H, Me–H), 2.25 (s, 12H, Me–H), 2.18 (s, 12H, Me–H) ppm; $^{13}$C NMR (126 MHz) (CDCl$_3$): $\delta$ = 169.2, 152.8, 138.9, 137.7, 137.7, 137.5, 135.8, 134.2, 132.9, 129.5, 128.1, 128.0, 127.3, 126.9, 124.7, 121.0, 21.3, 20.8, and 20.7 ppm; UV/Vis (CH$_2$Cl$_2$): $\lambda_{max}$ ($\varepsilon$[M$^{-1}$ cm$^{-1}$]) = 402 (132,023), 551 (48,426), 598 (20,366) nm; HRMS ($m/z$): [M]$^+$ calcd. for C$_{42}$H$_{36}$Br$_2$N$_5$, 1062.53; found 1062.49.

**Synthesis of 14**. **12** (20 mg, 0.019 mmol) was added to a round-bottomed 100 mL flask containing a magnetic bar and dissolved in CHCl$_3$/MeOH (15/6 mL). Pd (OAc)$_2$ (42.6 mg, 0.19 mmol) and NaOAc (18.7 mg, 0.22 mmol) was added, after being refluxed for 24 h, the solvent was evaporated in vacuo. The product was purified by column chromatography on silica-gel (CH$_2$Cl$_2$/n-hexane as an eluent). Recrystallization with n-hexane gave **14** (17.8 mg, 0.014 mmol, 73.6% yield) as green solids. $^1$H NMR (500 MHz) (CDCl$_3$): $\delta$ = 8.67 (d, 2H, $J$ = 1.8 Hz, C$_6$H$_3$-H), 7.59 (d, 2H, $J$ = 4.8 Hz, pyrrole-H), 7.31 (d, 2H, $J$ = 4.7 Hz, pyrrole-H), 7.25 (s, 2H, Ar-$m$-H), 7.03–7.01 (m, 6H, pyrrole-H and Ar-$m$-H), 6.98 (s, 2H, Ar-$m$-H), 6.94 (d, 2H, $J$ = 4.3 Hz, pyrrole-H), 6.67 (d, 2H, $J$ = 4.3 Hz, pyrrole-H), 6.59 (d, 2H, $J$ = 4.3 Hz, pyrrole-H), 5.32 (dd, 2H, $J$ = 7.6, 1.8 Hz, C$_6$H$_3$-H), 3.95 (d, 2H, $J$ = 7.7 Hz, C$_6$H$_3$-H), 2.44 (s, 6H, Me–H), 2.41 (s, 6H, Me–H), 2.31 (s, 6H, Me–H), 2.17 (s, 6H, Me–H), 2.16 (s, 6H, Me–H), 2.07 (s, 6H, Me–H); $^{13}$C NMR (126 MHz) (CDCl$_3$): $\delta$ = 172.4, 170.7, 170.6, 145.0, 144.2, 144.1, 140.5, 139.4, 139.2, 138.9, 137.8, 137.6, 137.6, 137.5, 137.4, 137.2, 136.7, 135.7, 133.9, 133.2, 132.6, 129.5, 128.0, 127.8, 127.7, 126.3, 126.2, 124.9, 123.7, 119.0, 21.2, 20.8, 20.7, 20.7, and 20.6 ppm; UV/Vis (CH$_2$Cl$_2$): $\lambda_{max}$ ($\varepsilon$[M$^{-1}$ cm$^{-1}$]) = 422 (73,384), 599 (17,949), 700 (21,503) nm; HRMS ($m/z$): [M]$^+$ calcd. for C$_{76}$H$_{62}$N$_6$Pd$_2$, 1270.3134; found 1270.3018.

**Synthesis of 13**. Route 1: A $p$-xylene-DMF solution (3/1.5 mL) of **9** (84.0 mg, 0.12 mmol), **11** (77.7 mg, 0.1 mmol), Pd$_2$(dba)$_3$ (18.5 mg, 0.02 mmol), PPh$_3$ (21.2 mg, 0.08 mmol), Cs$_2$CO$_3$ (71.0 mg, 0.2 mmol), and CsF (33.4 mg, 0.2 mmol) was degassed through three freeze-pump-thaw cycles, and the reaction Schlenk tube was purged with argon. The resulting mixture was stirred at reflux for 48 h. The reaction mixture was diluted with CHCl$_3$, washed with water, and dried over anhydrous sodium sulfate. After the solvent was removed, DDQ (65.8 mg, 0.29 mmol) was added to the resulting mixture in CH$_2$Cl$_2$ stirring for another 8 h. The reaction mixture was passed through a short alumina column (CH$_2$Cl$_2$ as an eluent). Evaporation of the solvent followed by silica-gel column chromatography (CH$_2$Cl$_2$/n-hexane as an eluent) and recrystallization with CH$_2$Cl$_2$/MeOH afforded **13** as violet solids (9.6 mg, 0.009 mmol, 9.0% yield).

Route 2: A toluene-DMF solution (3/1.5 mL) of **9** (180 mg, 0.25 mmol), 2,6-dibromopyridine (50.0 mg, 0.21 mmol), Pd$_2$(dba)$_3$ (38.8 mg, 0.042 mmol), PPh$_3$ (44.2 mg, 0.17 mmol), Cs$_2$CO$_3$ (138.0 mg, 0.42 mmol), and CsF (69.5 mg, 0.45 mmol) was degassed through three freeze-pump-thaw cycles, and the reaction Schlenk tube was purged with argon. The resulting mixture was stirred at reflux for 48 h. The reaction mixture was diluted with CHCl$_3$, washed with water, and dried over anhydrous sodium sulfate. After the solvent was removed, DDQ (136.2 mg,

0.6 mmol) was added to the resulting mixture in CH$_2$Cl$_2$ and this reaction mixture was stirred for another 8 h. The reaction mixture was passed through a short alumina column (CH$_2$Cl$_2$ as an eluent). Evaporation of the solvent followed by silica-gel column chromatography (CH$_2$Cl$_2$/n-hexane as an eluent) and recrystallization with CH$_2$Cl$_2$/MeOH afforded **13** as violet solids (9.1 mg, 0.009 mmol, 8.1% yield). $^1$H NMR (500 MHz) (CDCl$_3$): $\delta$ = 13.43 (br, 2H, N–H), 7.63 (d, 4H, $J$ = 4.5 Hz, pyrrole-H), 7.41 (d, 4H, $J$ = 8.0 Hz, pyridine-H), 7.01–6.98 (m, 12H, pyrrole-H and Ar-$m$-H), 6.35 (s, 4H, pyrrole-H), 6.05 (t, 2H, $J$ = 8.0 Hz, pyridine-H), 2.40 (s, 12H, Me–H), 2.26 (s, 12H, Me–H), 2.18 (s, 12H, Me–H) ppm; $^{13}$C NMR (126 MHz) (CDCl$_3$): $\delta$ = 169.2, 153.1, 150.8, 139.3, 137.9, 137.6, 137.5, 137.4, 135.2, 134.2, 133.8, 128.2, 128.1, 125.8, 123.5, 121.3, 21.3, 21.0, and 20.7 ppm; UV/Vis (CH$_2$Cl$_2$): $\lambda_{max}$ ($\varepsilon$[M$^{-1}$ cm$^{-1}$]) = 344 (40,195), 406 (128,444), 555 (46,048), 610 (16,662) nm; HRMS ($m/z$): [M]$^+$ calcd. for C$_{74}$H$_{64}$N$_8$, 1064.5248; found 1064.5239.

**Synthesis of 15, 16, and 17**. **13** (19.1 mg, 0.018 mmol) was added to a round-bottomed 100 mL flask containing a magnetic bar, and dissolved in CHCl$_3$/MeOH (15/6 mL). Pd(OAc)$_2$ (40.5 mg, 0.18 mmol) and NaOAc (17.8 mg, 0.21 mmol) was added, after being refluxed for 24 h, the solvent was evaporated in vacuo. The residue was purified by column chromatography on silica-gel (CH$_2$Cl$_2$/n-hexane as an eluent), three fractions were obtained. The first fraction was recrystallized with CH$_2$Cl$_2$/CH$_3$OH to afford **17** (2.8 mg, 2.2 μmol, 12.0% yield) as green solids. The second fraction was recrystallized with CH$_2$Cl$_2$/n-hexane to afford **15** (4.58 mg, 3.6 μmol, 20.0% yield) as dark green solids. The third fraction was recrystallized with CH$_2$Cl$_2$/CH$_3$OH to afford **16** (13.7 mg, 10.8 μmol, 60.0% yield) as navy blue solids.

**15**: $^1$H NMR (500 MHz) (CDCl$_3$): $\delta$ = 8.00 (d, 2H, $J$ = 4.7 Hz, pyrrole-H), 7.43 (d, 2H, $J$ = 4.7 Hz, pyrrole-H), 7.34 (d, 2H, $J$ = 4.7 Hz, pyrrole-H), 7.06–7.02 (m, 6H, Ar-$m$-H), 6.98 (s, 2H, Ar-$m$-H), 6.94 (d, 2H, $J$ = 4.7 Hz, pyrrole-H), 6.74 (d, 2H, $J$ = 4.5 Hz, pyrrole-H), 6.65 (d, 2H, $J$ = 4.3 Hz, pyrrole-H), 5.46 (d, 2H, $J$ = 7.6 Hz, pyridine-H), 4.07 (d, 2H, $J$ = 7.6 Hz, pyridine-H), 2.44 (s, 6H, Me–H), 2.42 (s, 6H, Me–H), 2.31 (s, 6H, Me–H), 2.18 (s, 6H, Me–H), 2.13 (s, 6H, Me–H), 2.09 (s, 6H, Me–H) ppm; $^{13}$C NMR (126 MHz) (CDCl$_3$): $\delta$ = 171.8, 170.7, 162.4, 157.4, 146.2, 144.9, 144.7, 144.1, 141.7, 140.6, 139.6, 138.4, 138.1, 137.7, 137.6, 137.6, 137.3, 136.8, 136.4, 135.5, 134.0, 133.4, 129.2, 128.2, 128.2, 127.0, 125.6, 121.9, 120.5, 21.5, 21.1, 21.0, and 21.0 ppm; UV/Vis (CH$_2$Cl$_2$): $\lambda_{max}$ ($\varepsilon$[M$^{-1}$ cm$^{-1}$]) = 309 (33,094), 405 (55,309), 461 (38,258), 597 (23,944), 660 (20,273), 723 (22,148) nm; HRMS ($m/z$): [M]$^+$ calcd. for C$_{74}$H$_{60}$N$_8$Pd$_2$, 1272.3038; found 1272.2944.

**16**: $^1$H NMR (500 MHz) (CDCl$_3$): $\delta$ = 7.67 (d, 2H, $J$ = 8.0 Hz, pyridine-H), 7.14 (d, 2H, $J$ = 4.5 Hz, pyrrole-H), 7.03 (t, 1H, $J$ = 7.7 Hz, pyridine-H), 6.95–6.92 (m, 12H, Ar-$m$-H and pyrrole-H), 6.52 (d, 2H, $J$ = 4.9 Hz, pyrrole-H), 6.25 (d, 2H, $J$ = 4.3 Hz, pyrrole-H), 6.22 (d, 2H, $J$ = 4.5 Hz, pyrrole-H), 3.69 (s, 1H, pyridine-H), 2.37 (s, 6H, Me–H), 2.35 (s, 6H, Me–H), 2.21 (s, 6H, Me–H), 2.15 (s, 6H, Me–H), and 2.12 (s, 12H, Me–H) ppm; $^{13}$C NMR (126 MHz) (CDCl$_3$): $\delta$ = 175.9, 169.9, 160.9, 155.2, 149.3, 147.4, 146.6, 144.7, 143.0, 140.5, 139.9, 138.0, 137.8, 137.3, 137.2, 137.2, 137.1, 136.8, 135.0, 134.4, 134.2, 133.4, 128.1, 128.0, 127.9, 127.4, 127.0, 126.9, 123.2, 121.7, 21.3, 20.8, 20.6, 20.6, and 20.5 ppm; UV/Vis (CH$_2$Cl$_2$): $\lambda_{max}$ ($\varepsilon$[M$^{-1}$ cm$^{-1}$]) = 403 (51,227), 556 (10,987), 664 (15,863), 777 (11,779), 925 (4759) nm; HRMS ($m/z$): [M]$^+$ calcd. for C$_{74}$H$_{60}$N$_8$Pd$_2$, 1272.3038; found 1272.3063.

**17**: $^1$H NMR (500 MHz) (CDCl$_3$): $\delta$ = 7.74 (d, 2H, $J$ = 5.3 Hz, pyrrole-H), 7.44 (t, 2H, $J$ = 7.9 Hz, pyridine-H), 7.31–7.29 (m, 4H, pyridine-H), 6.98–6.93 (m, 10H, Ar-$m$-H and pyrrole-H), 6.71 (d, 2H, $J$ = 5.0 Hz, pyrrole-H), 6.26–6.24 (m, 4H, pyrrole-H), 6.01 (d, 2H, $J$ = 5.3 Hz, pyrrole-H), 2.38 (s, 6H, Me–H), 2.31 (s, 6H, Me–H), 2.15 (s, 6H, Me–H), 2.09 (s, 6H, Me–H), 2.07 (s, 6H, Me–H), 2.03 (s, 6H, Me–H); $^{13}$C NMR (126 MHz) (CDCl$_3$): $\delta$ = 161.2, 157.1, 156.2, 154.7, 147.0, 140.4, 139.4, 138.9, 138.0, 137.8, 136.8, 136.7, 134.8, 134.6, 134.5, 133.5, 131.8, 128.1, 128.0, 127.8, 127.7, 125.2, 124.6, 117.0, 116.9, 113.2, 112.1, 100.3, 90.4, 21.2, 21.1, 21.1, 21.0, 20.7, and 20.6 ppm; UV/Vis (CH$_2$Cl$_2$): $\lambda_{max}$ ($\varepsilon$ [M$^{-1}$ cm$^{-1}$]) = 332 (38,615), 444 (51,873), 651 (15,961), 697 (27,095) nm; HR-MS (MALDI-TOF-MS): HRMS ($m/z$): [M]$^+$ calcd. for C$_{74}$H$_{62}$N$_8$Pd$_2$, 1274.3195; found 1274.3141.

## Data availability

The X-ray crystallographic coordinates for structures reported in this study have been deposited at the Cambridge Crystallographic Data Centre (CCDC), under deposition numbers 1959823, 1959827, 1959825, 1959828, 1959830, and 1975890 (**12**, **13**, **14**, **15**, **16**, and **17**). These data can be obtained free of charge from The Cambridge Crystallographic Data Centre via www.ccdc.cam.ac.uk/data_request/cif. The authors declare that all other data supporting the findings of this study are available within the paper and its supplementary information files.

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

## Acknowledgements

The work at Hunan Normal University was supported by the National Natural Science Foundation of China. (Grant Nos. 21772036, 21602058, and 21702057), Science and Technology Planning Project of Hunan Province (2018TP1017), Scientific Research Fund of Hunan Provincial Education Department (19A331). We thank Dr. Prof. Hiromitsu Maeda in Ritsumeikan University for MALDI-TOF MS Analysis.

## Author contributions

J.S. designed and conducted the project. L.L., Z.H., F.Z., Y.L., and T. T performed the synthesis and characterization, and measured the optical and electrochemical properties. L.X., M.Z., and T.T performed X-ray diffraction analysis and DFT calculations. A.O. and J.S. prepared the manuscript.

## Competing interests

The authors declare no competing interests.
