## [Peer Review File · Nature Communications]

REVIEWER COMMENTS

Reviewer #1 (Remarks to the Author):

The manuscript "Benzene- and Pyridine-Incorporated Octaphyrins (1.1.0.0.1.1.0.0) With Remarkably Different Coordination Modes Toward Two Pd(II) Metals" submitted by Jianxin Song and co-workers is a significant addition to expanded carbaporphyrinoid chemistry. Namely authors have elaborated the efficient methodology to construct octaphyrins (1.1.0.0.1.1.0.0) embedding two benzene (12) or two pyridine (13) moieties. These molecules acquire dumbbell geometry (D_{2h} symmetry) with inverted six membered rings in the "trans" positions. Authors claim that the novel octaphyrins reveal the nonaromatic properties. Importantly the suitably prearranged geometries of these octaphyrins, which created two (NNNC) pockets, facilitated incorporation of two palladium(II) cations. Thus the insertion into benzene octaphyrins (1.1.0.0.1.1.0.0) 12 yielded the C_{2v} symmetric molecule containing two identical Pd(II)-(NNNC) subunits (14). The insertion outcome was more complex for pyridine octaphyrins (1.1.0.0.1.1.0.0) (13). Here the analogous C_{2v} Pd(II)-(NNNC) species was identified (15) to be accompanied, however, by the Cs Pd(II)-(NNNC) complex (16). Eventually the intriguing reversible 16 - 16 conversion in the presence of TFA was detected. The remarkable rearrangement of a pyridine octaphyrins (1.1.0.0.1.1.0.0) 13 skeleton was discovered as well. This process afforded the bis-Pd(II)-(NNNN) complex 17 bearing a transannular C-C bond between the pyrrole α -positions.

The usual techniques (including MS, NMR, UV-Vis electronic spectroscopy and electrochemical methods) have been applied to characterize the novel macrocyclic compounds. The appropriate crystal structure have been also included. The DFT studies have been carried as well. Altogether the presented results are very significant. I highly appreciate the importance of this work, which provides a remarkable route to explore novel nontrivial coordination modes in expanded porphyrins and carbaporphyrinoids. Consequently, I can eventually recommend publication of this contribution in Nature Communication under condition that the remarks outlined below will be properly addressed.

Mechanism

The mechanism of the peculiar reversible 15 \rightleftharpoons 16 conversion is expected to be addressed in detail. Evidently the process requires the Pd-C β \rightleftharpoons Pd-(C β -H) transformation. Accordingly, the specific intramolecular activation of the C β -H bond, presumably facilitated by pyridine N-protonation can be of importance here. Interestingly, one can readily notice, that using TFA-d will result in selective β -deuteration of pyridine moieties, nicely providing the additional insight into the process.

In addition, the nature of the intermediate containing presumably the Pd-(NCCC) and Pd-((CH)NNN) units can be considered using the DFT approach.

Of course, one can wonder if a postulate above intermediate can be trapped by ¹H NMR in the presence of TFA in very large excess.

Peculiar chemical shift - issue

Consistently in the manuscript authors describe the reported molecules as nonaromatic. Considering the nature of α, α' -incorporated benzene or pyridine units such an electronic structure seems to be rationally expected. Still, the analysis of the ¹H NMR spectra at ESI, substantiated by examination of detailed chemical shifts, raises some essential scientific issue, which requires the sound addressing. In particular, the "enormous" upfield relocation of the inner-H resonances and remarkable difference between "inner" and "outer" chemical shifts have need of reasonable explanations (aromaticity?, the specific arrangement of six-membered units?). Below only the selected examples, which attracted my attention, are given (the assignments "outer" vs "inner" reflects the location with respect to the macrocyclic ring and has been given by this reviewer).

a) Figure S4 12 m-phenylene (outer) 8.34 vs m-phenylene (inner) 5.63

b) Figure S6 13 m-pyridine (inner) 6.05

c) Figure S8 C6H3-H (inner) 5.32, C6H3-H (inner) 3.95, C6H3-H (outer) 8.67

d) Figure S10, Pyridine-H (inner)), 5.46, Pyridine-H (inner), 4.07 .

e) Figure S12 pyridine-H (inner) 3.69 (coordinated) pyridine-H (inner, outer???) 7.67 , 7.03.

I believe that the references 15, 18 (this contribution) and J. Am. Chem. Soc. 2014, 136, 4281-4286 will be useful in the further analysis.

Evidently, the detailed assignments 1H resonances are absolutely necessary to provide the satisfactory explanation.

Other points

The acceptable HRMS data have been not been given in the description of new compounds.

Reviewer #2 (Remarks to the Author):

This manuscript presents the synthesis of expanded porphyrins with either 1,3-phenylene or 2,6-pyridinylene linkers, together with an investigation of the palladium complexes of these macrocycles. This study has uncovered several fascinating and unexpected features, such as the formation of the remarkable twisted bis-palladium complex 17. It is interesting that the 1,3-phenylene-linked macrocycle 12 forms only one Pd₂ complex, 14, whereas the 2,6-pyridinylene-linked macrocycle 13 forms three complexes 15, 16 and 17. All of these compounds are thoroughly characterized by NMR, mass spectrometry, electrochemistry, UV-visible spectroscopy and single-crystal X-ray analysis. The work has been carried out to a high standard and the manuscript is well written. It is suitable for publication in Nature Communications after a few minor revisions.

(1) Chart 1: The structure of compound 7 appears to be incorrect; it is lacking a nitrogen atom.

(2) Page 3, line 68: Why does the equimolecular reaction of 9 and 10 yield only a trace amount of 12 whereas 1.2 equivalents 9 produced 12 in 8.0% yield? Is this a reproducible result? Can the authors offer a plausible rationalization?

(3) Scheme 1. Replace "X = C" with "X = CH" (3 times).

(4) Page 11, lines 224-231: Were redox potentials measured on a CHI900 scanning electrochemical microscope or on an ALS660 electrochemical analyzer?

(5) Pages 11-14, lines 240, 260, 277, 288, 327 and 338. The quantity of DDQ should be specified. It is not adequate just to write "excess"; e.g. does this mean 1.1 equiv., 2 equiv. or 20 equiv.?

Reviewer #3 (Remarks to the Author):

The paper "Benzene- and Pyridine-Incorporated Octaphyrins(1.1.0.0.1.1.0.0) With Remarkably Different Coordination Modes Toward Two PdII Metals" by Liu et al. describes the synthesis of octaphyrins via Suzuki-Miyaura coupling. All compounds have been thoroughly characterized by NMR, SC-XRD, UV, CV and MALDI-TOF. Most SC-XRD data show however a rather bad agreement with the model. The disorders were just treated on a first level and these nice results demand and deserve a crystallographers love, e.g. for 13.cif the biggest residual densities belong to a 2nd site of the pyrrole group next to the pyridine ring and for 14.cif the phenylene unit is bent due to a not treated disorder of the phenylene group to mention just the most obvious mistakes. As there a two position for the palladium metal there have to be two positions for the phenylene unit. Instead of properly treating the disorder FLAT and ISOR was used to mask the problem. This strategy was also used in the figures of the manuscript where thermal ellipsoids are shown with 30% probability. Even though this study is overall well done, I do not recommend the publication of the manuscript in its current state.

Reviewer #4 (Remarks to the Author):

The manuscript brings interesting results on the title compounds. In addition to their syntheses the authors used various methods of their characterization. I am not an expert in organic syntheses and NMR experiments but this study is worth of publishing. My comments may be summarized as follows:

- i) It must be put more exactly (line 95) that "Complex 14 also shows a nearly C_{2h} symmetric structure ..." due to its deviations from planarity.
 - ii) The Table 1 description is a little bit chaotic. I propose to move a substantial part of lines 163-165 into the line 162.
 - iii) The DFT method use is mentioned in the line 200 and Figs. S50 – S51 but authors did not mention the software used. The references on the method and basis sets used are missing as well. Have the authors performed a geometry optimization of the structures under study (isolated or in solution?) or used X-ray geometries only (in such case – were the C-H and N-H bond lengths corrected and how the problem of disordered atoms has been solved)?
 - iv) Frontier MOs might be briefly mentioned in the main text as well (Figs. S40 – S44). It is a pity that the authors have not presented some results of population analysis for Pd atoms and their coordination polyhedra (at least in Supplementary). The manuscript would be stronger.
 - v) In Figs. S50-S51 the DFT energies of individual systems are compared. If the authors use optimized geometries, the Gibbs free energy data are more suitable for this purpose (temperature dependence).
 - vi) Discussion is too brief and descriptive only. Much deeper insight is desirable.
 - vii) In Method (lines 222-223) the X-ray experiment is described but the software used for the structure solution and related details are not mentioned at all.
 - viii) The asterisks in Figs. S2-S4, S6, S8, S10, S12 and S14 are not explained (impurities?).
 - ix) Some misprints and/or grammatical errors can be found by more careful reading.
- Finally it may be concluded that this manuscript demands major correction to be published.

Response to Comments of Reviewers

Response to Comments of Reviewer 1

Reviewer 1's general comments: The manuscript "Benzene- and Pyridine-Incorporated Octaphyrins (1.1.0.0.1.1.0.0) With Remarkably Different Coordination Modes Toward Two Pd(II) Metals" submitted by Jianxin Song and co-workers is a significant addition to expanded carbaporphyrinoid chemistry. Namely authors have elaborated the efficient methodology to construct octaphyrins (1.1.0.0.1.1.0.0) embedding two benzene (**12**) or two pyridine (**13**) moieties. These molecules acquire dumbbell geometry (D_{2h} symmetry) with inverted six membered rings in the "trans" positions. Authors claim that the novel octaphyrins reveal the nonaromatic properties. Importantly the suitably prearranged geometries of these octaphyrins, which created two (NNNC) pockets, facilitated incorporation of two palladium(II) cations. Thus the insertion into benzene octaphyrins (1.1.0.0.1.1.0.0) **12** yielded the C_{2v} symmetric molecule containing two identical Pd(II)-(NNNC) subunits (**14**). The insertion outcome was more complex for pyridine octaphyrins (1.1.0.0.1.1.0.0) (**13**). Here the analogous C_{2v} Pd(II)-(NNNC) species was identified (**15**) to be accompanied, however, by the C_s Pd(II)-(NNNC) complex (**16**). Eventually the intriguing reversible **16** – **16** conversion in the presence of TFA was detected. The remarkable rearrangement of a pyridine octaphyrins (1.1.0.0.1.1.0.0) **13** skeleton was discovered as well. This process afforded the bis-Pd(II)-(NNNN) complex **17** bearing a transannular C–C bond between the pyrrole α -positions. The usual techniques (including MS, NMR, UV-Vis electronic spectroscopy and electrochemical methods) have been applied to characterize the novel macrocyclic compounds. The appropriate crystal structures have been also included. The DFT studies have been carried as well. Altogether the presented results are very significant. I highly appreciate the importance of this work, which provides a remarkable route to explore novel nontrivial coordination modes in expanded porphyrins and carbaporphyrinoids. Consequently, I can eventually recommend publication of this contribution in Nature Communication under condition that the remarks outlined below will be properly addressed.

(1) Reviewer 1 wrote: Mechanism. The mechanism of the peculiar reversible **15** \rightleftharpoons **16**

conversion is expected to be addressed in detail. Evidently the process requires the $\text{Pd-C}_\beta \rightleftharpoons \text{Pd-(C}_\beta\text{-H)}$ transformation. Accordingly, the specific intramolecular activation of the $\text{C}_\beta\text{-H}$ bond, presumably facilitated by pyridine N-protonation can be of importance here. Interestingly, one can readily notice, that using TFA-d will result in selective β -deuteration of pyridine moieties, nicely providing the additional insight into the process.

In addition, the nature of the intermediate containing presumably the Pd-(NCCC) and Pd-((CH)NNN) units can be considered using the DFT approach. Of course, one can wonder if a postulate above intermediate can be trapped by $^1\text{H NMR}$ in the presence of TFA in very large excess.

Response: We examined the conversion from 15 to 16 with TFA-d by $^1\text{H NMR}$. In the figures below, the spectra (a) and (c) showed the $^1\text{H NMR}$ spectra of 15 and 16 in CDCl_3 , respectively. After addition of TFA-d to a solution of 15, the resulting mixture was washed with water to give the spectrum (b), which showed signals with similar chemical shifts and multiplets of 16 except those peaks ascribed to the proton c, indicating that C-H activation and the following C-Pd bonds deuteration underwent in the isomerization process. Unfortunately, we could not observe the intermediate *via in-situ* $^1\text{H NMR}$ but confirmed that 16 was a sole product. We appreciated the nice advice from the reviewer 1. Figure S55 and Figure S56 show the relative energies of these intermediates indicated by DFT calculation.

(2) **Reviewer 1 wrote: Peculiar chemical shift – issue.** Consistently in the manuscript authors

describe the reported molecules as nonaromatic. Considering the nature of α,α' -incorporated benzene or pyridine units such an electronic structure seems to be rationally expected. Still, the analysis of the ^1H NMR spectra at ESI, substantiated by examination of detailed chemical shifts, raises some essential scientific issue, which requires the sound addressing. In particular, the “enormous” upfield relocation of the inner-H resonances and remarkable difference between “inner” and “outer” chemical shifts have need of reasonable explanations (aromaticity?, the specific arrangement of six-membered units?). Below only the selected examples, which attracted my attention, are given (the assignments “outer” vs “inner” reflects the location with respect to the macrocyclic ring and has been given by this reviewer).

a) Figure S4 **12** m-phenylene (outer) 8.34 vs m-phenylene (inner) 5.63

b) Figure S6 **13** m-pyridine (inner) 6.05

c) Figure S8 $\text{C}_6\text{H}_3\text{-H}$ (inner) 5.32, $\text{C}_6\text{H}_3\text{-H}$ (inner) 3.95, $\text{C}_6\text{H}_3\text{-H}$ (outer) 8.67

d) Figure S10 Pyridine-H (inner) 5.46, Pyridine-H (inner), 4.07.

e) Figure S12 pyridine-H (inner) 3.69 (coordinated) pyridine-H (inner, outer???) 7.67, 7.03.

I believe that the references 15, 18 (this contribution) and J. Am. Chem. Soc. 2014, 136, 4281-4286 will be useful in the further analysis.

Evidently, the detailed assignments ^1H resonances are absolutely necessary to provide the satisfactory explanation.

Response: Chemical shifts of the peripheral protons at the skeleton of the octaphyrins gave us the information on aromaticity. In judgement of the aromaticity the octaphyrins, the chemical shifts of all the peripheral protons should be considered as a whole. Apparently, the facts that the pyrrolic NH protons in 12 and 13 were observed at around 13 ppm indicates that they are not aromatic and the chemical shifts of the pyrrolic β -protons were observed at 6.0-7.5 ppm, supporting the above conclusion. For compounds 14, 15 and 16, the chemical shifts of the pyrrolic β -protons indicate that they are not aromatic.

Although all of the compounds 12-16 are not aromatic, as reviewer 1 referred, the differences in chemical shifts between outer H and inner H in phenylene/ C_6H_3 /pyridine units are large. This phenomenon can be rationalized by considering the magnetic field induced by the local ring current of phenylene/ C_6H_3 /pyridylene units. In 12 the inner H of m-phenylene lies above the center of the opposite m-phenylene moiety, which means they are

in the shielding area. However, the outer H of *m*-phenylene is far from the center of the opposite *m*-phenylene, which means it is not in the shielding area or even in the deshielding area. As a result, the chemical shifts of the inner and outer protons differ significantly. As the molecule of 12 is symmetric, this interpretation is also appropriate for the protons in the other *m*-phenylene moiety.

Since the compounds 13, 14 and 15 share a similar molecular geometry, similar phenomenon was observed in these molecules. One point to be addressed here is that the distance between the *m*-phenylene units in 14 is shorter than that in 12 due to the coordination of Pd atoms. Consequently, the differences in chemical shifts between inner and outer protons in 14 is much larger than that in 12. Similarly, the chemical shifts of the inner pyridine protons in 15 are more upfield shifted than those in 13. (All of these protons in 13 and 15 show smaller chemical shifts than those 'normal' pyridine-protons.)

For compound 16, the case is different from those in 12-15 due to their different symmetries. Apparently in 16 the chemical environment of C₅NH unit differs from that of C₅NH₃ unit. The protons of the C₅NH unit lies above the center of the C₅NH₃ unit, which means this proton locates in the shielding area. However, the 3 protons of the C₅NH₃ unit are not so close to the center of the C₆H unit, which indicates these 3 protons are less shielded. As a result, the proton of the C₅NH shows a chemical shift of 3.69 ppm, while the protons of C₅NH₃ show chemical shifts of 7.67 and 7.03 ppm.

As suggested by reviewer 1 and to confirm our explanation above, DFT calculation was applied to elucidate these chemical shifts. These results were summarized in Table S8-S13, which indicated that the theoretical calculation met well with the experimental data. Sharp contrast between the centers of these molecules and the centers of hemiporphyrins in NICS value support our previous judgment that these octaphyrins are not global aromatic and that 'abnormal' chemical shifts are resulted from local aromaticity.

(3) **Reviewer 1 wrote:** The acceptable HRMS data have been not been given in the description of new compounds.

Response: We repeated all of the HRMS experiments, acceptable HRMS data were included in the main text.

Response to Comments of Reviewer 2

Reviewer 2's general comments: This manuscript presents the synthesis of expanded porphyrins with either 1,3-phenylene or 2,6-pyridinylene linkers, together with an investigation of the palladium complexes of these macrocycles. This study has uncovered several fascinating and unexpected features, such as the formation of the remarkable twisted bis-palladium complex **17**. It is interesting that the 1,3-phenylene-linked macrocycle **12** forms only one Pd₂ complex, **14**, whereas the 2,6-pyridinylene-linked macrocycle **13** forms three complexes **15**, **16** and **17**. All of these compounds are thoroughly characterized by NMR, mass spectrometry, electrochemistry, UV-visible spectroscopy and single-crystal X-ray analysis. The work has been carried out to a high standard and the manuscript is well written. It is suitable for publication in Nature Communications after a few minor revisions.

(1) **Reviewer 2 wrote:** Chart 1: The structure of compound **7** appears to be incorrect; it is lacking a nitrogen atom.

Response: Corrected.

(2) **Reviewer 2 wrote:** Page 3, line 68: Why does the equimolecular reaction of **9** and **10** yield only a trace amount of **12** whereas 1.2 equivalents **9** produced **12** in 8.0% yield? Is this a reproducible result? Can the authors offer a plausible rationalization?

Response: We repeated this experiment for several times. The yield of the equimolecular reaction of 9 and 10 was confirmed to be 3.8%. The relatively low yield of the equimolecular reaction may be due to some impurities in 9 or chemical reactivity of 9 under this cross coupling condition. Ref. (Angew. Chem. Int. Ed. 2016, 55, 6438; 2019, 58, 8124)

(3) **Reviewer 2 wrote:** Scheme 1. Replace “X = C” with “X = CH” (3 times).

Response: Corrected.

(4) **Reviewer 2 wrote:** Page 11, lines 224-231: Were redox potentials measured on a CHI900 scanning electrochemical microscope or on an ALS660 electrochemical analyzer?

Response: We used ALS660 electrochemical analyzer to measure redox potentials, this was clarified in *Materials and characterization* part.

(5) **Reviewer 2 wrote:** Pages 11-14, lines 240, 260, 277, 288, 327 and 338. The quantity of DDQ should be specified. It is not adequate just to write “excess”; e.g. does this mean 1.1 equiv., 2

equiv. or 20 equiv.?

Response: A typical quantity of DDQ we applied is 2.4 equiv. The quantity was noted in the manuscript.

Response to Comments of Reviewer 3

Reviewer 3's general comments: The paper "Benzene- and Pyridine-Incorporated Octaphyrins(1.1.0.0.1.1.0.0) With Remarkably Different Coordination Modes Toward Two PdII Metals" by Liu et al. describes the synthesis of octaphyrins via Suzuki-Miyaura coupling. All compounds have been thoroughly characterized by NMR, SC-XRD, UV, CV and MALDI-TOF.

(1) Reviewer 3 wrote: Most SC-XRD data show however a rather bad agreement with the model. The disorders were just treated on a first level and these nice results demand and deserve a crystallographers love, e.g. for 13.cif the biggest residual densities belong to a 2nd site of the pyrole group next to the pyridine ring and for 14.cif the phenylene unit is bent due to a not treated disorder of the phenylene group to mention just the most obvious mistakes. As there are two positions for the palladium metal there have to be two positions for the phenylene unit. Instead of properly treating the disorder FLAT and ISOR was used to mask the problem. This strategy was also used in the figures of the manuscript where thermal ellipsoids are shown with 30% probability. Even though this study is overall well done, I do not recommend the publication of the manuscript in its current state.

Response: We thank the suggestion of Reviewer 3. As Reviewer 3 referred serious disorders were observed in 12, 13, 14 and 15. We split all of the atoms in 12 but did not treat 13, 14 and 15 in a similar way initially. Inspired by Reviewer 3 we re-refined crystallographic data of 13, 14 and 15 by trying to find the 2nd parts in these compounds. For 13, the R1 value decreased significantly when the 2nd positions were found and treated as the 2nd part despite large amount of constrains and restrains involving AFIX, FLAT, SADI, ISOR and DELU were used. When this strategy was applied for 15, the 2nd part could be isolate as well but the R1 value did not decrease. In 14 we could find only one 2nd part of a half molecule in an asymmetric unit (one asymmetric unit contains 2 halves of molecule), however, the structure is not stable any more during refining when the 2nd part of the other half molecule was

isolated. This is mainly due to the low occupancy (~0.15) of the 2nd part.

Response to Comments of Reviewer 4

Reviewer 4's general comments: The manuscript brings interesting results on the title compounds. In addition to their syntheses the authors used various methods of their characterization. I am not an expert in organic syntheses and NMR experiments but this study is worth of publishing. My comments may be summarized as follows:

Finally it may be concluded that this manuscript demands major correction to be published.

(1) Reviewer 4 wrote: It must be put more exactly (line 95) that “Complex **14** also shows a nearly C_{2h} symmetric structure ...” due to its deviations from planarity.

Response: We rephrased this sentence as “Complex **14** shows a nearly C_i symmetric structure ...”

(2) Reviewer 4 wrote: The Table 1 description is a little bit chaotic. I propose to move a substantial part of lines 163-165 into the line 162.

Response: Changed.

(3) Reviewer 4 wrote: The DFT method use is mentioned in the line 200 and Figs. S50 – S51 but authors did not mention the software used. The references on the method and basis sets used are missing as well. Have the authors performed a geometry optimization of the structures under study (isolated or in solution?) or used X-ray geometries only (in such case – were the C-H and N-H bond lengths corrected and how the problem of disordered atoms has been solved)?

Response: All calculations were carried out using the *Gaussian 09* program. Initial geometries for 12-17 were obtained from X-ray structures. The structures were fully optimized without any symmetry restriction. Geometry optimizations in the ground state (S₀) were performed by the density functional theory (DFT) method with restricted B3LYP (Becke's three-parameter hybrid exchange functionals and the Lee-Yang-Parr correlation functional) level employing a basis set of 6-311G(d,p) for C, H, N and SDD for Pd.^[S3] NICS values were calculated with GIAO method at the B3LYP level employing the same basis sets for geometry optimizations. Calculated chemical shifts were estimated relative to the magnetic shielding of a proton of chloroform (24.95 ppm) calculated at the same level. These

details for DFT calculation were added in SI.

(4) Reviewer 4 wrote: Frontier MOs might be briefly mentioned in the main text as well (Figs. S40 – S44). It is a pity that the authors have not presented some results of population analysis for Pd atoms and their coordination polyhedra (at least in Supplementary). The manuscript would be stronger.

Response: MOs were shown in Figure S51 – S53 and description of MOs was added to the main text (Page 6). The Mulliken atomic charge values are illustrated in Figure S54 and the coordination polyhedra of Pd atoms are shown in Figure S47 – S50.

(5) Reviewer 4 wrote: In Figs. S50-S51 the DFT energies of individual systems are compared. If the authors use optimized geometries, the Gibbs free energy data are more suitable for this purpose (temperature dependence).

Response: Gibbs free energy data were added in Figure S55 and Figure S56 and the corresponding data was discussed in main text.

(6) Reviewer 4 wrote: Discussion is too brief and descriptive only. Much deeper insight is desirable.

Response: We some analysis in ¹H NMR spectra was added in SI (Table S8 to S13).

(7) Reviewer 4 wrote: In Method (lines 222-223) the X-ray experiment is described but the software used for the structure solution and related details are not mentioned at all.

Response: Using Olex2, structures of compound 12-17 were solved with the ShelXS structure solution program using Direct Methods and refined with the ShelXL refinement package using Least Squares minimisation. Disordered solvent molecules were treated by SQUEEZE program of Platon. This description was added in Method part.

(8) Reviewer 4 wrote: The asterisks in Figs. S2-S4, S6, S8, S10, S12 and S14 are not explained (impurities?).

Response: The asterisks in Figs. S2-S4, S6, S8, S10, S12 and S14 indicate impurities. They are mainly CHCl₃ and H₂O in CDCl₃, CH₂Cl₂ and grease. Details were noted in these Figure.

(9) Reviewer 4 wrote: Some misprints and/or grammatical errors can be found by more careful reading.

Response: Some misprints and/or grammatical errors were corrected, they are highlighted in the revised version.

REVIEWERS' COMMENTS

Reviewer #1 (Remarks to the Author):

Authors have satisfactorily addressed all previously raised problems, consequently I can recommend this contribution for publication.

Reviewer #2 (Remarks to the Author):

The revised version of this manuscript has been significantly improved and all the points raised by the referees have been addressed. This is a fascinating piece of work and it has been carried out to a high standard.

Two minor points:

(1) In their response to Reviewer 1's first point, the authors have carried out an interesting study of the site of deuteration on treatment of compound 15 with TFA-d to form deuterated 16. These mechanistic results should be mentioned in the text and added to the SI.

(2) It is not very clear what is meant by "relative energy" in Figures S55 and S56 and why this is different from " ΔG ". Should "relative energy" be changed to " ΔH "?

Reviewer #4 (Remarks to the Author):

The manuscript brings interesting information on the title compounds but its presentation must be improved. My comments are related to its Supplementary information only and can be summarized as follows:

- i) The title and authors are missing.
- ii) How did you check the stability of the optimized structures (l. 408)?
- iii) Missing reference of 6-311G(D,p) basis sets (l. 411).
- iv) SDD denotes both pseudopotentials and basis sets (l. 412).
- v) Some details on NICS treatment are desirable at l. 412 (probe location – NICS(0) or NICS(1)?).

Reviewer #5 (Remarks to the Author):

I have been requested to re-examine the updated crystallography. In general the authors have now appropriately dealt with the previously raised issues (disorder models) and the structures are now acceptable. The authors have also done a good job of dealing with the comments of the other referees and the manuscript is not largely acceptable.

In the future the authors should consider using the newer versions of shelxl for their refinements which have a significantly improved way of dealing with refinements of structures that have been squeezed.

Response to Comments of Reviewers.

Thank you very much for your work for handling this revised manuscript.

Response to Comments of Reviewers

Response to Comments of Reviewer 1

Reviewer 1's general comments: Authors have satisfactorily addressed all previously raised problems, consequently I can recommend this contribution for publication.

Response: We thank the evaluation of Reviewer 1.

Response to Comments of Reviewer 2

Reviewer 2's general comments: The revised version of this manuscript has been significantly improved and all the points raised by the referees have been addressed. This is a fascinating piece of work and it has been carried out to a high standard.

(1) **Reviewer 2 wrote:** In their response to Reviewer 1's first point, the authors have carried out an interesting study of the site of deuteration on treatment of compound **15** with TFA-d to form deuterated **16**. These mechanistic results should be mentioned in the text and added to the SI.

Response: These mechanistic results are mentioned in the text and added to the SI.

(2) **Reviewer 2 wrote:** It is not very clear what is meant by "relative energy" in Figures S55 and S56 and why this is different from "delta G". Should "relative energy" be changed to "delta H"?

Response: The "relative energy" is the difference in their electronic energies including zero-point-energy correction (E+ZPE). In the revised manuscript, the explanation has been added, and both delta H and delta G values are listed.

Response to Comments of Reviewer 4

Reviewer 4's general comments: The manuscript brings interesting information on the title compounds but its presentation must be improved. My comments are related to its Supplementary information only and can be summarized as follows:

(1) **Reviewer 4 wrote:** The title and authors are missing.

Response: The title and authors are added in the first page of Supplementary information.

(2) **Reviewer 4 wrote:** How did you check the stability of the optimized structures (l. 408)?

Response: We used the crystal structures as the initial geometries and optimized it based on the structure. For each optimized structure, frequency calculations were conducted to confirm no imaginary frequency. Given the closed shell characters probed by experimental results, we did not consider any open shell characters for calculations.

(3) **Reviewer 4 wrote:** Missing reference of 6-311G(D,p) basis sets (l. 411).

Response: A suitable reference has been added: Krishnan, R., Binkley, J. S., Seeger, R., Pople, J. A. Self-consistent molecular orbital methods. XX. A basis set for correlated wave functions. *J. Chem. Phys.* **72**, 650-654 (1980)

(4) **Reviewer 4 wrote:** SDD denotes both pseudopotentials and basis sets (l. 412).

Response: The corresponding statements were corrected.

(5) **Reviewer 4 wrote:** Some details on NICS treatment are desirable at l. 412 (probe location – NICS(0) or NICS(1)?)

Response: All the NICS calculation means NICS(0) values because the molecules are non-planar and some are three-dimensional. The definition of z axis is not obvious and we are afraid that it may confuse the readers' understanding. In the revised manuscript, all the description NICS has been corrected as NICS(0).

Response to Comments of Reviewer 5

Reviewer 4's general comments: I have been requested to re-examine the updated crystallography. In general the authors have now appropriately dealt with the previously raised issues (disorder models) and the structures are now acceptable. The authors have also done a good job of dealing with the comments of the other referees and the manuscript is not largely acceptable. In the future the authors should consider using the newer versions of shelxl for their refinements which have a significantly improved way of dealing with refinements of structures that have been squeezed.

Response: We thank Reviewer 5's work in re-examining updated crystallography and the beneficial suggestion of using newer versions of shelxl in future. Probably Reviewer 5 means that this manuscript is largely acceptable according to the context.